# A Simulation Case Study of Knee Joint Compressive Stress during the Stance Phase in Severe Knee Osteoarthritis Using Finite Element Method

**DOI:** 10.3390/medicina57060550

**Published:** 2021-05-30

**Authors:** Takashi Fukaya, Hirotaka Mutsuzaki, Toshiyuki Aoyama, Kunihiro Watanabe, Koichi Mori

**Affiliations:** 1Department of Physical Therapy, Faculty of Health Sciences, Tsukuba International University, 6-8-33 Manabe, Tsuchiura 300-0051, Japan; 2Center for Medical Sciences, Ibaraki Prefectural University of Health Sciences, 4669-2 Ami-machi, Inashiki 300-0394, Japan; mutsuzaki@ipu.ac.jp; 3Department of Physical Therapy, Ibaraki Prefectural University of Health Sciences, 4669-2 Ami, Inashiki 300-0394, Japan; aoyamato@ipu.ac.jp; 4Department of Radiology, Shin-Oyama City Hospital, 2251-1 Hitotonoya, Oyama, Tochigi 323-0827, Japan; k.watanabe@hospital.oyama.tochigi.jp; 5Department of Radiological Sciences, Ibaraki Prefectural University of Health Sciences, 4669-2 Ami-machi, Inashiki 300-0394, Japan; mori@ipu.ac.jp

**Keywords:** knee osteoarthritis, finite element method, musculoskeletal model, compressive stress distribution, ground reaction force

## Abstract

*Background and Objectives*: Medial knee osteoarthritis is known to increase the mechanical load on the medial compartment of the knee joint during walking; however, it is not visually understood how much the mechanical load increases nor where in the medial compartment of the knee joint that load is focused. Therefore, we conducted a simulation study to determine the location and amount of the mechanical load in the medial compartment of the knee joint during the stance phase. *Materials and Methods*: Subject was a patient with right medial knee osteoarthritis. Computed tomography imaging and gait analysis were performed on subject. The CT image of the right knee was calculated using finite element analysis software. Since this software can set the flexion angle arbitrarily while maintaining the nonuniform material properties of the bone region, the model is constructed by matching the knee joint extension image obtained by CT to the loading response phase of gait analysis. The data of muscle exertion tension and vertical ground reaction force were inserted into the knee joint model created from the computed tomography-based finite element method, and the knee joint compressive stress was calculated. *Results*: With regard to compressive stress, the tibia showed high stress at 4.10 to 5.36 N/mm^2^. The femur showed high stress at 4.00 to 6.48 N/mm^2^. The joint compressive stress on the medial compartment of the knee joint was found to concentrate on the edge of the medial tibial condyle in the medial knee osteoarthritis subject. *Conclusions*: The measurement method of knee joint compressive stress by computed tomography-based finite element method can visually be a reliable method of measuring joint compressive stress in the medial knee osteoarthritis. This reflects the clinical findings because concentration of stress on the medial knee joint was observed at the medial osteophyte.

## 1. Introduction

The onset of knee osteoarthritis (KOA) alters load sharing between the medial and lateral compartments of the tibiofemoral joint [1]. These changes result in pain, which is the most common complaint of patients with KOA [2,3]. Medial KOA with higher severity on X-rays, depicted by narrowing of the knee joint space and formation of osteophytes, enlarges the knee varus angle and increases the load on the medial compartment of the knee joint. Because medial knee load cannot be directly measured in vivo, the external knee adduction moment during the stance phase is used to assess the dynamic medial knee load [4]. The external knee adduction moment in medial KOA appears as varus deformation [5] and causes dysfunction associated with the mechanical load on the knee joint and degeneration of cartilage [6]. In addition, patients with medial KOA have varus thrust; that is, they have an increased varus angle characteristically seen during the initial stance phase during walking [7]. Varus thrust presents as a range of motion limitation and muscle weakness in medial KOA [8,9], and it increases both the external knee adduction moment and the mechanical loads in the medial compartment [10]. The medial KOA rehabilitation strategy attempts to reduce the mechanical load on the knee joint by inserting insoles and strengthening the quadriceps muscles. Therefore, identifying the characteristics of the mechanical load is important in considering rehabilitation strategies to reduce the mechanical load in order to improve knee joint function and walking ability. Medial KOA increases the mechanical load on the medial compartment of the knee joint; however, it is unclear exactly which area of the medial compartment experiences the increased load. We have previously reported the stress on the knee joint by healthy subjects using FEM from the results of gait analysis, but have not reported on the case of KOA [11]. In this report, we used a developed measurement method to calculate knee joint stress with muscle exertion tension and the vertical value of ground reaction force using computed tomography-based finite element method (CT-FEM), and we examined which part of the knee medial compartment of the KOA received the most mechanical load at the loading response phase during walking. We hypothesized that in medial KOA with varus deformity, the mechanical load at the loading response phase during walking would be concentrated in the narrow medial area.

## 2. Materials and Methods

### 2.1. Subject

Subject was a female patient with medial KOA (Age: 77 years old, Height: 1.44 m, Weight: 53 kg). The medial KOA subject had a clinical history that included Kellgren–Lawrence (KL) grades [12] and had bilateral KOA with four levels of KL classification. We performed CT imaging and gait analysis on subject. The subject’s KL classification was of equal severity on both sides. However, the Numerical Rating Scale for pain was 6 on the right and 5 on the left. The femoro-tibial angle was 187° on the right and 184° on the left. The right knee was used for analysis because the right knee was scheduled for total knee arthroplasty. Subject provided written informed consent, and the protocol for this study was approved by the Ibaraki Prefectural University of Health Sciences Ethics Committee (e98).

### 2.2. Gait Analysis

Participant changed into tight-fitting shorts, removed footwear, and walked barefoot along a 5 m-long walkway at a self-selected speed (0.96 m/s). Kinematic data were acquired using a 3D motion analysis system (Vicon Nexus, Oxford, UK) with eight cameras operating at 200 Hz. Two force platforms (Kistler Instruments, Winterthur, Switzerland) embedded in the laboratory floor captured ground reaction forces at 1200 Hz synchronized with the kinematic data. Before measurement, height, weight, lower limb length (from anterior superior iliac spines to medial malleolus), width of anterior superior iliac spines, knee width, and foot width of participants were measured. According to a lower extremity model of the Plug-In-Gait^®^ marker set (Vicon, Oxford Metrics Group) [13], reflective markers were placed on the following bilateral anatomical landmarks: anterior and posterior superior iliac spines, lateral thighs, lateral femoral epicondyles, lateral calves, lateral malleoli, calcanei, and the base of the second metatarsals on the dorsal aspect of the feet. The gait parameters during the stance phase, determined from the force plate data, were normalized to the values at 100% during the stance phase (foot-strike to toe-off = 100%) using spline interpolation. The stance phase of the gait was defined as five periods [14]: initial contact (0% of the stance: IC), loading response (0~16% of the stance: LR), mid-stance (17~50% of the stance: MS), terminal stance (51~83% of the stance: TS), and preswing (84~100% of the stance: PS). Kinematic and ground reaction force data were filtered using a Butterworth low pass filter with 6 and 30 Hz cut-off frequencies, respectively.

### 2.3. Musculoskeletal Modelling

Muscle exertion tensions were estimated using the musculoskeletal model by AnyBody (Anybody Modeling System, AnyBody Technology, Denmark) [15]. Based on the marker data obtained from the gait analysis, the segment length and the joint angle was estimated. AnyBody outputted the optimized segment length and motion angle for each part and performed kinematic analysis. Thereafter, size scaling was performed by applying the human body data of the AnyBody system to the calculated size of the subject, and the segment length and the center of the joint were calculated [16]. Further, by adding the ground reaction force data and executing the optimization algorithm, muscle strength, which are the forces inside the body, are calculated. In this study, the estimation of muscle exertion tension around the knee joint were set as follows: quadriceps femoris, biceps femoris, semimembranosus, semitendinosus, and gracilis muscles.

### 2.4. Computed Tomography-Based Finite Element Method (CT-FEM)

The CT image of the right knee was calculated using MECHANICAL FINDER Extended Edition (Research Centre of Computation Mechanics, Tokyo, Japan), which is finite element analysis software. Since this software can set the flexion angle arbitrarily while maintaining the nonuniform material properties of the bone region, the model is constructed by matching the knee joint extension image obtained by CT to LR phase of gait analysis. The image calculation range was from the distal end of the femur, about 10 cm above the knee joint, to the distal end of the tibia and fibula. In the subject’s knee joint model, joint components such as ligaments, meniscus, and articular cartilage were created based on material properties and reflected in CT images [11]. The quadriceps femoris, biceps femoris, semimembranosus, semitendinosus, and gracilis muscles based on AnyBody were set as the muscles that exert tension around the knee joint. The ligaments included the patellar, anterior cruciate, posterior cruciate, medial collateral, and lateral collateral ligaments. In the construction of FEM, the bone part was used as a solid element of the tetrahedral element. In addition, a 1.0-mm-thick triangular prism shell element was placed to cover the surface of the patella. These values adopted the recommended size based on previous studies [11,17,18]. For the Poisson’s ratio of the shell element, 0.4, which is a constant value peculiar to the bone material, was used [19,20]. The length of one side was about 1.0 to 2.0 mm. Keyak’s conversion formula was used for the material properties of the bone element [21,22,23]. In this conversion formula, a linear relationship is established between the CT value and the bone density, and the bone density is calculated from the CT value. Three-dimensional modeling software Metasequoia (Tetraface, Inc., Tokyo, Japan) was used to model soft tissue elements, menisci, ligaments, and cartilage. The ligaments use truss elements (wire elements) to not become rigid against bending or compression [11]. The constraint condition for this computational model was that the proximal end of the femur was completely constrained. Furthermore, at the distal ends of the tibia and fibula, the horizontal direction (defined as the X and Y axes) was constrained. The reaction force against the sum of the force plate data and each muscle force is the joint reaction force. Therefore, only the vertical component (defined as the Z axis) was calculated. As a result, this computational model evaluated only the force plate data in the *Z*-axis direction [11]. Because the quadriceps can cause improper deformation due to the tension of the truss elements, we used the forced displacement method for quadriceps traction. The amount of displacement was 3 mm. Contact conditions were set for the femur and meniscus, the femur and tibia, the tibia and meniscus, and the cartilage between the femur and patella. The coefficient of friction in the contact analysis was 0.01. The lateral edge of the medial meniscus and the lower end of the tibial cartilage were connected to avoid slipping of the meniscus under load. In medial KOA cases, much of the medial meniscus had disappeared, and only its lateral edge was reflected in the model.

## 3. Results

Figure 1 showed X-rays and MRI images. X-rays showed narrowing of the medial space, osteosclerosis of the subchondral bone, and widening of the osteophytes. MRI images also revealed degeneration of the medial meniscus.

This section may be divided by subheadings. It should provide a concise and precise description of the experimental results, their interpretation, as well as the experimental conclusions that can be drawn.

Table 1 showed the data of knee joint angles and vertical component of ground reaction force at LR during the stance phase, which were substituted in the analysis by CT-FEM.

Figure 2 showed the right knee joint taken during the operation of the subject. The white arrows indicate that the cartilage on the medial femoral joint surface and tibial joint surface has was completely destroyed and bone was exposed. The red arrow points the degenerated and torn medial meniscus. From the results in Figure 1 and Figure 2, this was a shape very similar to the meniscus created by the CT-FEM model (Figure 3).

It was found that joint compressive stress was concentrated on the edge of the medial tibial condyle in the KOA subject (Figure 3 and Figure 4). With regard to compressive stress, in the tibia of Figure 4, the red part showed high stress at 4.10 to 5.36 N/mm^2^, and the blue and green parts showed low stress at 1.01 to 3.18 N/mm^2^. It was concentrated at the edge of the medial tibial condyle in the KOA subject, and, in the femur of Figure 4, the red part showed high stress at 4.00 to 6.48 N/mm^2^, and the blue and green parts showed low stress at 1.02 to 3.57 N/mm^2^. The higher stress concentration was also found with medial femoral condyle in the KOA subject.

The left figure shows the articular surface of the tibia, and the right figure shows that of the femur.

## 4. Discussion

In past reports, some studies on the joint stress of the knee joint have been conducted [24,25], but the analysis of the joint stress by CT-FEM of the knee joint based on the data obtained by the gait analysis of KOA has been not reported. This study showed that by using CT-FEM, a developed measurement method of knee joint stress which inputs muscle exertion tension and ground reaction force, compressive stress was concentrated at the edge of the medial tibial condyle in a patient with severe medial KOA. In severe medial KOA, collapse of joint structure due to varus deformation occurs, and it is assumed that external knee adduction moment increases the mechanical load on the medial tibial condyle [4]. From the findings of the knee joint taken during subject’s surgery, X-rays and MRI images, the shape of the meniscus is almost distorted, and it is difficult to disperse the mechanical load. Li et al. [26] reported that FEM analysis showed that greater meniscal damage could lead to greater joint stress and progression to KOA. In the case of this study as well, the medial meniscus was severely damaged, and the tibia had increased joint stress along the medial margin. From the X-rays of this case, varus deformity is progressing, and it is inferred that the external knee adduction moment during walking increases. In addition, MRI images showed that the medial meniscus was degenerated due to the narrowing of the medial joint space. It was considered that the formation of osteophytes on the medial margin of the knee joint caused lateral thrust during the stance phase of walking, and the load on the tibial joint surface was biased toward the medial margin. Chantarapanich et al. [27] reported that varus knee joints have higher stresses in the medial compartments while normal knee joints have higher stresses in the lateral compartments. As a result of FEM in the case of this study, the stress of the medial femoral condyle became high, and the result is considered to be valid from the past reports and the intraoperative photographs of this case.

Increased knee joint loading is a detrimental factor in the progression of KOA [28], and increased varus thrust and external knee adduction moment cause overload on the knee joint that deviates from normal. In particular, varus thrust associated with varus deformity is a factor that accelerates the progression of KOA [28], and should be the target of intervention [29]. Therefore, preventive intervention that delays the progression of OA before deformation occurs is considered important. In this study, we targeted the most severe patient, and it is predicted that the local load on the knee joint will increase as the severity increases. In the future, it will be necessary to perform CT-FEM analysis on mild to moderate patients and perform a detailed evaluation of the load on the knee joint.

The developed measurement method of knee joint compressive stress by CT-FEM can visually be a reliable method of measuring joint compressive stress in the medial KOA subject. This reflects the clinical findings because concentration of stress on the medial knee joint was observed at the medial osteophyte. In this study, we examined the joint stress of the knee joint at the LR period during stance phase, when the load on the joint is the largest in the walking. Since varus thrust occurs during this period and the joint force also increases, it is considered that the stress increased at the site different from the past reports [25,26]. From the results of knee joint stress in this study, it is inferred that varus thrust and joint force of knee joint at the LR period are key factors for the progression of KOA.

There are several limitations to this study. The participants in this study was a female with severe KOA. Since the measurement was performed by one subject, the results of this study cannot be generalized. This study is a model in which only the LR period during stance phase is extracted. Therefore, it does not reflect all of KOA’s varus deformities. We need to recruit an adequate number of participants to validate the results of this study.

This study showed that in severe KOA, failure of the joint structure due to a decrease in muscle strength and varus deformation increases the mechanical load on the medial side of the knee joint. As a result, it may be causing pain in related tissues around the medial side of the knee joint. The results of the CT-FEM are clinically relevant, and further studies could be undertaken to improve the accuracy of the measurement of data, analysis other than stance phases during walking, and investigate for the prevention of KOA progression.

## Figures and Tables

**Figure 1 medicina-57-00550-f001:**
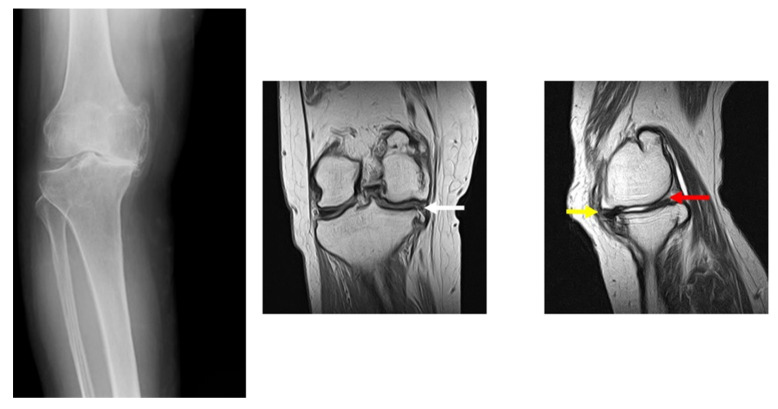
X-rays and MRI images of KOA. X-rays was performed at the load position. MRI shows the results of the coronal plane on the right side and the sagittal plane on the left side. White arrows deviate from the medial meniscus and indicate degenerative rupture. The yellow arrow indicates degeneration of the medial meniscus anterior segment, and the red arrow indicates degeneration rupture of the medial meniscus posterior segment.

**Figure 2 medicina-57-00550-f002:**
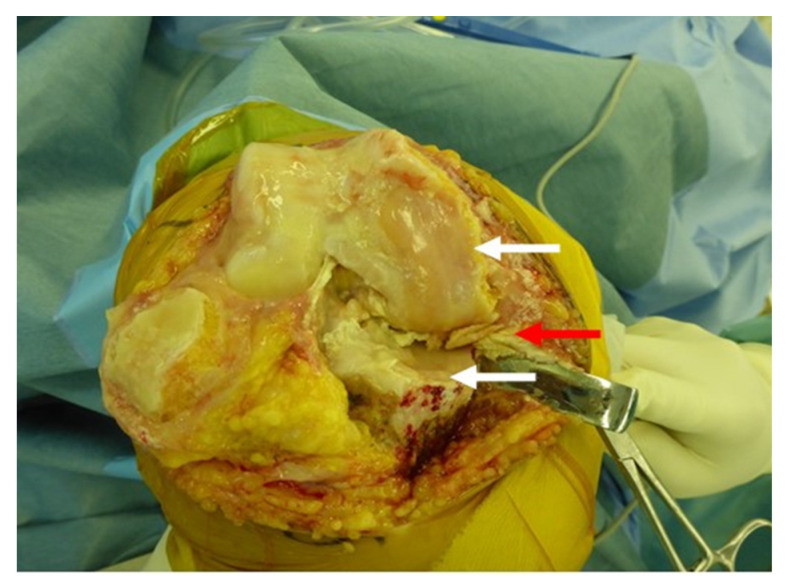
Right knee joint removed during Subject’s surgery. The white arrows indicate that the cartilage on the medial femoral joint surface and tibial joint surface was completely destroyed and bone was exposed. The red arrow points to the degenerated and torn medial meniscus.

**Figure 3 medicina-57-00550-f003:**
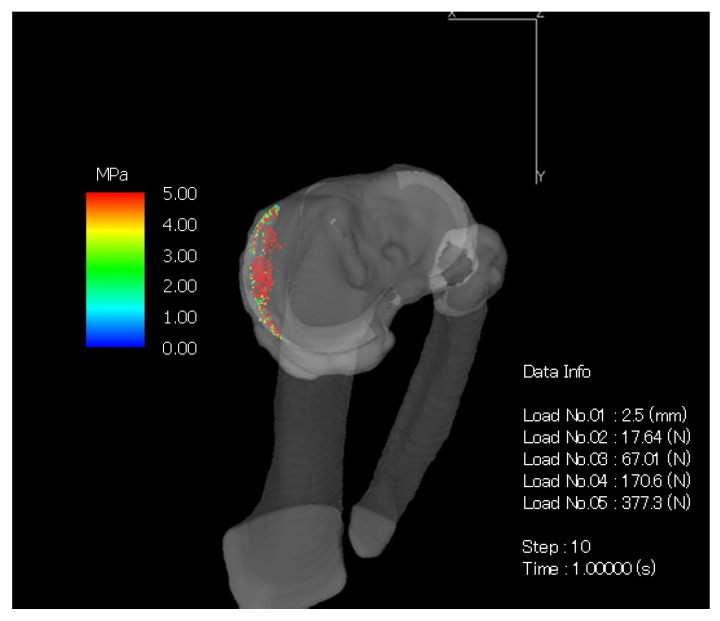
Compression stress using computed tomography-based finite element method (CT-FEM) representing the shape of the meniscus.

**Figure 4 medicina-57-00550-f004:**
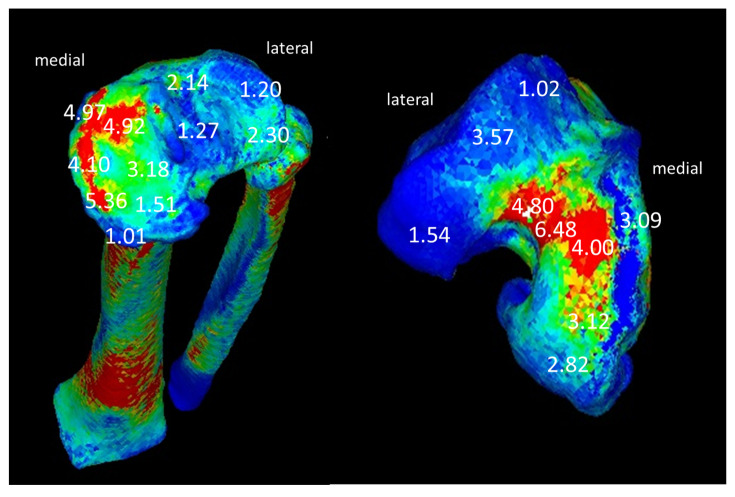
Compression stress using CT-FEM on the articulating surface of the femur and tibia.

**Table 1 medicina-57-00550-t001:** Knee joint angles and vertical component of ground reaction force.

	Knee Joint Angles (Degrees)			Vertical Component of Ground Reaction Force (N)
	Flexion	Adduction	External Rotation	
KOA Subject	0.3	16.2	11.9	371.9

## Data Availability

All the data are available from the corresponding author upon reasonable request.

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
