# Peer review of "A Simulation Case Study of Knee Joint Compressive Stress during the Stance Phase in Severe Knee Osteoarthritis Using Finite Element Method"

_medicina, 2021, doi:10.3390/medicina57060550_

Round 1

Reviewer 1 Report

The paper present a knee joint analysis using CT-FEM methodology.

The paper i s very well structured in terms of formal analysis, methodology description and results.

The paper has some minor changes that should be addressed.

In the discussion there should be made more emphasis on joint sane behavior and going deeper to the potential of your proposal.

The quality of figure 3 and 4 should be improved the number can be read with difficulties and it is the key part of your proposal.

Reviewer 2 Report

In reviewed manuscript authors present a case study of a patient with medial knee osteoarthritis who underwent experimental diagnostic for joint compressive stress evaluation. Manuscript may be considered for publication after minor changes:

Material and Methods:

  • Line 83: there was only one participant, so change “participants”; please look also to line 239;

Results:

  • Table 2 is blunted: there are more lines than text (only one row contains data) – it’s better to place this data in text
  • Figure 2:
  • line 172 there should be arrows (2 arrows on photo) and “indicate”;
  • line 173 red arrow should be “points” not “is”; joint surface was not “disappeared” but was completely destroyed and bone was exposed ; please rephrase this;
  • description to figure 2 : not “taken” but rather “removed” or “replaced”
  • line 187 space between figure 4.

Discussion:

  • Lines 217-219: this sentence is confusing and must be rephrased;
  • Line 226 should be “patient”
